# Can fMRI reveal the representation of syntactic structure in the brain?

**Aniketh Janardhan Reddy**
Machine Learning Department
Carnegie Mellon University
ajreddy@cs.cmu.edu

**Leila Wehbe**
Machine Learning Department
Carnegie Mellon University
lwehbe@cmu.edu

## Abstract

While studying semantics in the brain, neuroscientists use two approaches. One is to identify areas that are correlated with semantic processing load. Another is to find areas that are predicted by the semantic representation of the stimulus words. However, most studies of syntax have focused only on identifying areas correlated with syntactic processing load. One possible reason for this discrepancy is that representing syntactic structure in an embedding space such that it can be used to model brain activity is a non-trivial computational problem. Another possible reason is that it is unclear if the low signal-to-noise ratio of neuroimaging tools such as functional Magnetic Resonance Imaging (fMRI) can allow us to reveal the correlates of complex (and perhaps subtle) syntactic representations. In this study, we propose novel multi-dimensional features that encode information about the syntactic structure of sentences. Using these features and fMRI recordings of participants reading a natural text, we model the brain representation of syntax. First, we find that our syntactic structure-based features explain additional variance in the brain activity of various parts of the language system, even after controlling for complexity metrics that capture processing load. At the same time, we see that regions well-predicted by syntactic features are distributed in the language system and are not distinguishable from those processing semantics. Our code and data will be available at https://github.com/anikethjr/brain_syntactic_representations.

## 1 Introduction

Neuroscientists have long been interested in how the brain processes syntax. To date, there is no consensus on which brain regions are involved in processing it. Classically, only a small number of regions in the left hemisphere were thought to be involved in language processing. More recently, the language system was proposed to involve a set of brain regions spanning the left and right hemisphere [1]. Similarly, some findings show that syntax is constrained to specific brain regions [2, 3], while other findings show syntax is distributed throughout the language system [4–6].

The biological basis of syntax was first explored through studies of the impact of brain lesions on language comprehension or production [7] and later through non-invasive neuroimaging experiments that record brain activity while subjects perform language tasks, using methods such as functional Magnetic Resonance Imaging (fMRI) or electroencephalography (EEG). These experiments usually isolate syntactic processing by contrasting the activity between a difficult syntactic condition and an easier one and by identifying regions that increase in activity with syntactic effort [3]. An example of these conditions is reading a sentence with an object-relative clause (e.g. "The rat *that the cat chased* was tired"), which is more taxing than reading a sentence with a subject-relative clause (e.g. "The cat *that chased the rat* was tired"). In the past decade, this approach was extended to study syntactic processing in naturalistic settings such as when reading or listening to a story [8–11]. Because such complex material is not organized into conditions, neuroscientists have instead devised

35th Conference on Neural Information Processing Systems (NeurIPS 2021).

complexity metrics capturing the word-by-word evolving syntactic demands required to understand the material. Using these metrics, neuroscientists were able to identify regions with activity correlated with syntactic processing load, and suggest the involvement of these regions in syntactic processing.

While many have studied syntactic processing as captured through complexity measures, very few have studied the syntactic representations themselves. As an analogy, consider the neurobiology of language semantics. A large part of the literature has focused on characterizing the semantic processing load related to integrating or predicting incoming words (see [12] for a review). Concurrently, another part has focused on studying where the meaning of words itself is represented, either through contrast-based studies (see [13] for a review) or through encoding model approaches (e.g. [14–19]). It can be argued however, that in syntax, most research focuses on the first approach (using complexity) and not the second (using representations).

There are two main reasons why the study of syntactic representations using fMRI is difficult:

**1.** The first reason is computational. To identify the brain correlates of syntactic representations, one has to embed the syntactic representation (often a tree) into a vector that can be then used to predict the time series of brain activity as a function of syntactic structure. This vector representation should change as the words are processed incrementally. The construction of such a vector space is akin to the problem of building a graph embedding, and it is not trivial. The goal is to have different sentences or segments of sentences with similar structure (irrespective of their meanings) map to nearby points in the vector space.

**2.** The second reason is that the fMRI signal is noisy, and it is not clear that the neural basis of the representation of syntactic structure lends itself to being studied using fMRI. For instance, it could be that the neural substrate for syntax is intermingled with that of other language components like semantics and is hard to disentangle, or it could be that neurons inside the same voxel perform different syntactic computations, making it hard to differentiate the signal corresponding to these different computations. More generally, it could also be that syntactic computations are organized in a way that the low Signal-to-Noise Ratio (SNR) of fMRI makes it difficult to study them.

To help address the first difficulty, we propose syntactic structure embeddings that encode the syntactic information inherent in natural text that subjects read in the scanner. These structure embeddings are proposed as an additional tool in the arsenal of neurolinguists and can serve to ask the question of whether the fMRI signal can reveal syntactic representations. We use our syntactic structure embeddings in a voxelwise encoding model framework [14, 16, 17, 20]. We find that our embeddings – along with other simpler syntactic structure embeddings built using conventional syntactic features such as part-of-speech (POS) tags and dependency role (DEP) tags – are able to explain an additional portion of the variance in the fMRI data of subjects reading text, even after controlling for complexity metrics that capture processing load, suggesting that fMRI can indeed reveal syntactic representations. Furthermore, the regions that are well-predicted by syntactic representations are distributed across the language network.

We also attempt to minimize the amount of semantic information present in our embeddings. After showing that they do not encode a significant amount of semantics, we use our syntactic embeddings to address the aforementioned issue of whether regions that are predicted by syntactic features are selective for syntax, meaning they are only responsive to syntax and not to other language properties such as semantics. To answer this question, we use a contextual word embedding space [21] that integrates semantics and syntax. Consistent with prior literature, regions that are predicted by syntax are much better predicted by the contextual embeddings and do not appear to be selective for syntax.

## 2   Background

**Naturalistic tasks**    Naturalistic tasks are gradually becoming the paradigm of choice to study language processing in the brain. Many have used naturalistic listening tasks to investigate syntactic processing by measuring word-by-word processing load [8–10, 22–25]. Even though story reading or listening involves multiple types of processing (low-level processing, word recognition, building sentences, inferring meaning and social reasoning, etc.), researchers are gradually learning more and more about how these processes are mapped to different brain areas. It has already been shown that the regions defined by Fedorenko et al. [26] as the language network (also used in this work) process word-level syntactic [16–18] and semantic features as well as sequence meaning [19, 27, 28]. Processing information beyond the level of sentences is thought to recruit other areas [17, 29].

Encoding models are frequently used to analyze data collected using naturalistic tasks. These are models that are trained to predict brain activity at every voxel as a function of some feature of interest. Voxels that are well-predicted are thought to be involved in processing the information encoded in the feature. The various complexity metrics were formulated as features that encode syntactic demands.

**Complexity metrics** Several complexity metrics have been proposed to capture word-by-word syntactic demands. One common approach for constructing a syntactic complexity metric is to assume a sentence's syntactic representation and estimate the number of syntactic operations performed at each word. Node Count is a popular metric constructed using this approach [30]. It relies on constituency trees (structures that capture the hierarchical grammatical relationship between the words in a sentence). While traversing the words of the sentence in order, subtrees of this tree get completed; Node Count refers to the number of such subtrees that get completed at each word, effectively capturing syntactic load or effort. Important work in neurolinguistics [8] has used Node Count to support the theory that the Anterior Temporal Lobe (ATL) is involved in syntactic processing.

Complexity metrics are a good proposal for identifying brain regions involved in processing the syntactic structure of a sentence. They can be used to study specific hypotheses about how syntactic structure is built. For example, one could compute these metrics using different types of parsers to determine which parser's operations are more similar to the brain's structure building processes [22]. Using these metrics also leads to more robust results when compared to those obtained by contrasting syntactic conditions. This is because of two reasons. First, the naturalistic stimuli used in such studies are very diverse. Second, since the metrics can be computed at every word (i.e. they are incremental metrics), the correlation between syntactic processing load and the recorded brain activity can be robustly estimated using a large number of data points (i.e. words).

**Representing syntactic structure** The focus of this work is to study syntactic representations in the brain by building features that encode a sentence's syntactic structure and using them in voxelwise encoding models. Constituency trees and dependency trees are the two main structures that capture a sentence's syntactic structure. Constituency trees are derived using phrase structure grammars (PSGs) that encode valid phrase and clause structure (see fig. 1(A) for an example). Dependency trees encode relations between pairs of words such as subject-verb relationships. We derive representations from both types of trees. We derive word-level DEP tags from dependency trees, and encode the structural information given by constituency trees to capture the hierarchical representations of phrase structure. We characterize the syntactic structure inherent in sentence constituency trees by computing an evolving vector representation of the syntactic structure processed at each word using the subgraph embedding algorithm by Adhikari et al. [31].

We resort to developing this new embedding pipeline because we want our constituency tree-based embeddings to be almost purely syntactic. One might suggest using the internal representations of a neural parser (such as the parser used by Hale et al. [9]) to directly encode syntax. However, neural parsers use word embeddings, making it likely for there to be some semantics in the internal representations. This "leakage" of semantics makes it very difficult to isolate the brain regions that process syntax. The problem is further compounded by previous findings which indicate that syntax and semantics might be processed in the same regions of the brain [6], making it impossible to infer whether a region's activity is correlated with the syntax or the semantics encoded in the features.

## 3 Methods

We first describe the syntactic features used in this study and their generation. All of the features we use are incremental i.e. they are computed per word. We then describe our fMRI data analyses.

**Complexity metrics** We use four complexity metrics in our analyses - Node Count, Syntactic Surprisal, word frequency and word length. To compute Node Count, we obtain the constituency tree of each sentence using the self-attentive encoder-based constituency parser by Kitaev and Klein [32]. We compute Node Count for each word as the number of subtrees that are completed by incorporating this word into its sentence. Syntactic Surprisal is another complexity metric proposed by Roark et al. [33] and is computed using an incremental top-down parser [34]. It measures how unexpected it is to read a given word in the current syntactic context. Both of these metrics aim to measure the amount of effort that is required to integrate a word into the syntactic structure of its sentence. The word frequency metric is computed using the wordfreq package [35] as the Zipf frequency of a word. This is the base-10 logarithm of the number of occurrences per billion of a given word in a large text

corpus. Finally, word length is the number of characters in the presented word. The last two metrics approximate the amount of effort that is required to read a word, irrespective of its context.

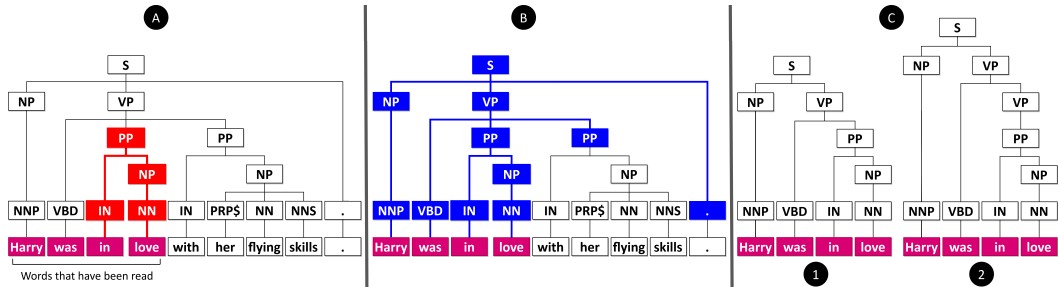

Figure 1: Example of complete and incomplete subtrees and two possible partial parses: Part A shows a sentence's constituency tree generated by a self-attentive encoder-based constituency parser [32] using all of its words. The largest completed subtree for "love" is highlighted in red. The incomplete subtree generated till "love" is highlighted in blue in part B. Incomplete subtrees are generally much deeper than complete ones. In part C, we can see two possible partial parses generated by an incremental top-down parser [34] only using the words till "love". We see that the structure of the two parses is slightly different. The supplementary material contains a document illustrating complete and incomplete subtrees for words from a long sentence.

**Constituency Tree-based Graph Embeddings (ConTreGE)**   Constituency trees are a rich source of syntactic information. We build three representations of these trees that encode this information:

**1.** The largest subtree which is completed upon incorporating a word into a sentence (see fig. 1(A)) is representative of the implicit syntactic information given by the word. A subtree is considered complete when all of its leaves are terminals. The largest subtree completed by a given word refers to the subtree with the largest height that also satisfies the following conditions - the given word must be one of its leaves and all of its leaves must only contain words that have been seen till then. Given that Node Count reduces all of the information present in these subtrees to just one number, it is easy to see that it cannot fully capture this information. POS tags (categorize words into nouns, verbs, adjectives, etc.) also capture some of the information present in these trees as they encode lexical syntactic information. But, they are incapable of completely encoding their hierarchical structure and the parsing decisions made while generating them. To better encode their structure, we first build subgraph embeddings of these completed subtrees called ConTreGE Comp vectors.

**2.** We hypothesize that the brain not only processes structure seen thus far but also predicts future structure from structure it already knows. To test this, we construct embeddings, called ConTreGE Incomp vectors, using incomplete subtrees that are constructed by retaining all the PSG productions that are required to derive the words seen till then, starting from the root of the sentence's tree. A non-terminal node is expanded only if it eventually leads to the derivation of a word that has been seen. Any other non-terminal nodes are not expanded. Encoding such subtrees allows us to capture future sentence structure (in the form of future constituents) before the full sentence is read (see fig. 1(B)). These subtrees contain leaves that are non-terminal symbols unlike complete subtrees that only have terminal symbols (words and punctuation) as leaves. In this context, a non-terminal symbol is a symbol that can be derived further using some rule in the PSG (e.g. NP, VP, etc.). If incomplete subtrees are more representative of the brain's processes, it would mean that the brain correctly predicts certain phrase structures even before the entire phrase or sentence is read. ConTreGE Comp and ConTreGE Incomp vectors need to be built using accurate constituency trees constructed using the whole sentence. Thus, we reuse the trees generated to compute Node Count to build them.

**3.** Further, the brain could be computing several possible top-down partial parses that can derive the words seen thus far (see fig. 1(C)) and modifying the list of possible parses as future words are read. To test this hypothesis, we designed Incremental ConTreGE (InConTreGE) vectors that are representative of the most probable parses so far. For a given word, its InConTreGE vector is computed as: $v = \sum_{i=1}^{5} e^{-s_i} W_i$ where $W_i$ is the subgraph embedding of a partial parse tree built by an incremental top-down parser [34] after reading the word and $s_i$ is the score assigned to this partial parse that is inversely proportional to the parser's confidence in this tree. The InConTreGE feature space is constructed to encode the different possible parse trees that can derive the words seen so far. These different parse trees try to encode uncertainty about the parent nodes of the words seen so

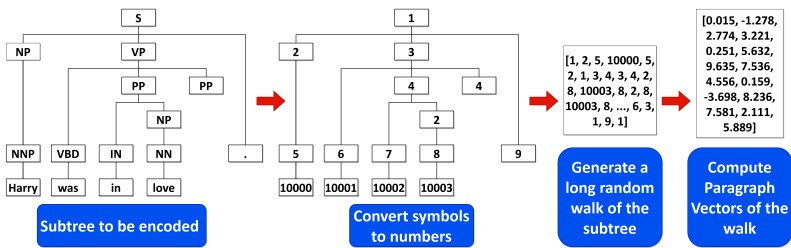

Figure 2: Steps for encoding subtrees.

far. They also do not encode any information about possible future children of these parent nodes. However, with ConTreGE Incomp, we encode parts of the final parse tree of a sentence. It does not encode any uncertainty. With this embedding space we are trying to measure the ability of the brain to correctly predict future constituents. We see uncertainty about the current parse and the ability to correctly predict future constituents as two related but distinct concepts. This is the reason why we use the ConTreGE Incomp to hypothesize about the brain encoding correct future information.

To effectively capture the structure of all subtrees, we encode them using the subgraph embeddings proposed by Adhikari et al. [31] which preserve the neighbourhood properties of subgraphs. A long fixed length random walk on a subgraph is generated to compute its embedding. Since consecutive nodes in a random walk are neighbours, a long walk can inform us about the neighbourhoods of nodes in the subgraph. Each node in a walk is identified using its unique ID. So, a random walk can be interpreted as a "paragraph" where the words are the node IDs. Finally, the subgraph's embedding is computed as the Paragraph Vector [36] of this paragraph that is representative of the subgraph's structure. Note that all of the subtrees of a given type (complete, incomplete or partial parse) are encoded together, ensuring that all ConTreGE Comp, all ConTreGE Incomp and all InConTreGE vectors are in their own spaces. Fig. 2 illustrates the subtree encoding process. First, every unique non-terminal in the subtrees is mapped to a unique number (e.g. S is mapped to 1, NP is mapped to 2, etc.) and every terminal is mapped to a unique number that is representative of the order in which they were presented (the first presented token is mapped to 10000, the second token is mapped to 10001 and so on). We did not map each unique terminal to a unique number (for instance, we did not map all instances of "Harry" to one number) because a random walk through the tree could give us word co-occurrence information and thus lead to the inclusion of some semantics in the vectors.

Every tree node's label is then replaced by the number it was mapped to in the previous step. The edge lists of these subtrees are supplied to the subgraph embedding generation algorithm to finally obtain 15-dimensional vectors for every presented word. The length of the random walks is set to 100000 and we use an extension of the Distributed Bag of Words (DBOW) model proposed by Le and Mikolov [36] for generating Paragraph Vectors called Sub2Vec-DBON by Adhikari et al. [31]. The sliding window length is set to 5 and the model is trained for 20 epochs. Since ConTreGE Comp, ConTreGE Incomp and InConTreGE encode information about the neighbourhoods of all nodes in the constituency trees, they can capture their hierarchical structure. Thus, brain regions predicted by these vectors are likely to be involved in building and encoding hierarchical sentence structure.

**Punctuation**    We create one-hot binary vectors indicating the type of punctuation that was presented along with a word (e.g. **.** or **,**). For example, a sentence might have ended with "Malfoy.". In this punctuation-based feature space, the column corresponding to **.** will be set to 1 for this word. While punctuation is seldom considered a syntactic feature, sentence boundaries are highly correlated with changes in working memory load. These changes are bound to be a great source of variability in the fMRI signal (as we will observe later). Failing to account for sentence boundaries and working memory might be a source of confounding that has been ignored in the literature.

**Part-of-speech tags and dependency tags**    We use two standard word-level syntactic features - POS and DEP tags. The POS tag of a word is read off previously generated constituency trees (those obtained using the Kitaev and Klein [32] parser). The DEP tag of a word (e.g. subject, object, etc.) corresponds to its assigned role in the dependency trees of the presented sentences which were generated using the spaCy English dependency parser [37]. We create one-hot binary vectors indicating the POS tag and the DEP tag of each word and concatenate them to create one feature space which we refer to as simple syntactic structure embeddings.

**Semantic features**    We adapt the vectors obtained from layer 12 of a pretrained [38] cased BERT-large model [21] to identify regions that process semantics. We use layer 12 because of previous work

showing that middle layers of sentence encoders are optimal for predicting brain activity [27, 28]. To compute the semantic embedding of a word, we first feed the word and the words that precede it in its sentence to the pretrained model, preventing the inclusion of future semantic information. Then, we average the embeddings of all the input words and use this average to represent the semantic information seen till then. Using principal component analysis (PCA), we reduce the dimensionality of the average to 15 to match the ConTreGE vectors' dimensionality and to also reduce overfitting (the BERT embeddings have 1024 dimensions and we only have 1291 time points in the fMRI data).

**fMRI data**    We use the fMRI data of 9 subjects reading chapter 9 of *Harry Potter and the Sorcerer's Stone* [39], collected and made available freely without restrictions by Wehbe et al. [17]. Participants gave their written informed consent and the study was approved by the Carnegie Mellon University Institutional Review Board [17]. Words were presented serially for 0.5s each. All brain plots shown here are averages over the 9 subjects in the Montreal Neurological Institute (MNI) space. Further details are in Appendix A.

**Predicting brain activity**    The applicability of a given syntactic feature in studying syntactic processing is determined by its efficacy in predicting the brain data described above. Ridge regression is used to perform these predictions and their coefficient of determination ($R^2$ score) measures the feature's efficacy. For each voxel of each subject, the regularization parameter is chosen independently. We use ridge regression because of its computational efficiency and because of the Wehbe et al. [40] results showing that with such fMRI data, as long as the regularization parameter is chosen by cross-validation for each voxel independently, different regularization techniques lead to similar results. Indeed, ridge regression is a common regularization technique used for predictive fMRI models [41, 17, 16]. These models that predict brain activity are called voxelwise encoding models.

For every voxel, a model is fit to predict the signals $Y = [y_1, y_2, \ldots, y_n]$ recorded in that voxel where $n$ is the number of time points (TR, or time to repetition). The words are first grouped by the TR in which they were presented. Then, features of words in every group are summed to form a sequence of features $X = [x_1, x_2, \ldots, x_n]$ aligned with the brain signals. The response measured by fMRI is an indirect consequence of neural activity peaking about 6 seconds after stimulus onset. A common way to account for this delay is to express brain activity as a function of the features of the preceding time points [41, 17, 16]. Thus, we train our models to predict any $y_i$ using $x_{i-1}, x_{i-2}, x_{i-3}$ and $x_{i-4}$.

We test the models in a cross-validation loop: the data is first split into 4 contiguous and equal sized folds. Each model uses three folds of the data for training and one fold for evaluation. We remove the data from the 5 TRs which either precede or follow the test fold from the training set of folds. This is done to avoid any unintentional data leaks since consecutive $y_i$s are correlated with each other because of the lag and continuous nature of the fMRI signal. The brain signals and the word features which comprise the training and testing data for each model are individually Z-scored. After training we obtain the predictions for the validation fold. The predictions for all folds are concatenated (to form a prediction for the entire experiment in which each time point is predicted from a model trained without that time point). Note that since all 3 ConTreGE vectors are stochastic, we construct them 5 times each, and learn a different model each time. The 5 model predictions are averaged together into a single prediction. The $R^2$ score is computed at each voxel using the predicted and real signals.

We run a permutation test to test if $R^2$ scores are significantly higher than chance. We permute blocks of contiguous fMRI TRs, instead of individual TRs, to account for the slowness of the underlying hemodynamic response. We choose a common value of 10 TRs [18]. The predictions are permuted within fold 5000 times, and the resulting $R^2$ scores are used as an empirical distribution of chance performance, from which the p-value of the unpermuted performance is estimated. We also run a bootstrap test to test if a model has a higher $R^2$ score than another. The difference is that in each iteration, we sample with replacement the predictions of both models for a block of TRs and compute the difference of their $R^2$ and use the resulting distribution to estimate the p-value of the unpermuted difference. Finally, the Benjamni-Hochberg False Discovery Rate correction [42] is used for all tests (appropriate because fMRI data is considered to have positive dependence [43]). The correction is performed by grouping together all the voxel-level $p$-values (i.e. across all subjects and feature groups) and choosing one threshold for all of our results. The correction is done in this way as we test multiple prediction models across multiple voxels and subjects. To compute Region of Interest (ROI) statistics, left-hemisphere ROI masks for the language system obtained from a "sentence vs. non-word" fMRI contrast [26] are obtained from [44] and mirrored to obtain right-hemisphere ROIs.

# 4   Results

We use different sets of feature spaces as inputs to encoding models that predict activity in each fMRI voxel. Many of our feature spaces have overlapping information - POS and DEP tags include punctuation, BERT vectors have been shown to encode syntax [45] and ConTreGE vectors, built from constituency trees, encode some POS tags information. To detect brain regions sensitive to the distinct information given by a feature space, we build hierarchical feature groups in increasing order of syntactic information and test for significant differences in prediction performance between two consecutive groups. We start with the simplest feature – punctuation, and then add more complex features in order: the complexity metrics, POS and DEP tags, one of the ConTreGE vectors and the vectors derived from BERT (which can be thought of as a super-set of semantics and syntax). At each step, we test if the introduction of the new feature space leads to significantly larger than chance improvement in $R^2$ (similar to [46]). Fig. 3 and 4 summarize our results (Appendix C has the raw prediction results and Appendix D contains detailed plots for every ROI showing the subject-specific values).

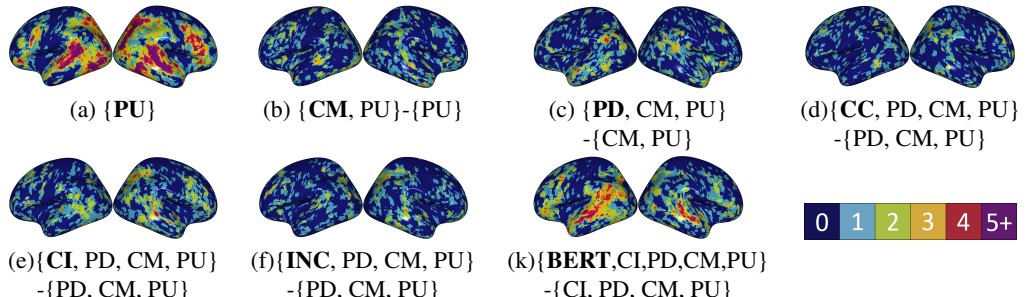

(a) {**PU**}  (b) {**CM**, PU}-{PU}  (c) {**PD**, CM, PU} -{CM, PU}  (d){**CC**, PD, CM, PU} -{PD, CM, PU}

(e){**CI**, PD, CM, PU} -{PD, CM, PU}  (f){**INC**, PD, CM, PU} -{PD, CM, PU}  (k){**BERT**,CI,PD,CM,PU} -{CI, PD, CM, PU}

Figure 3: The first plot shows the number of subjects for which a given voxel is significantly predicted by punctuation ($p \leq 0.05$). The others show the number of subjects for which the difference in $R^2$ scores between two feature groups is significant ($p \leq 0.05$). Here, PU = Punctuation, CM = All complexity metrics, PD = POS and DEP Tags, CC = ConTreGE Comp, CI = ConTreGE Incomp, INC = InConTreGE, BERT = BERT embeddings and '{,}' indicates that these features were concatenated in order to make the predictions. '-' indicates a hypothesis test for the difference in $R^2$ scores between the two feature groups being larger than 0. We see that the distinct information given by syntactic structure-based features does explain additional variance in the brain activity, even after controlling for the complexity metrics and punctuation. The semantic vectors are also very predictive and many well-predicted regions overlap with those that are predicted by syntax.

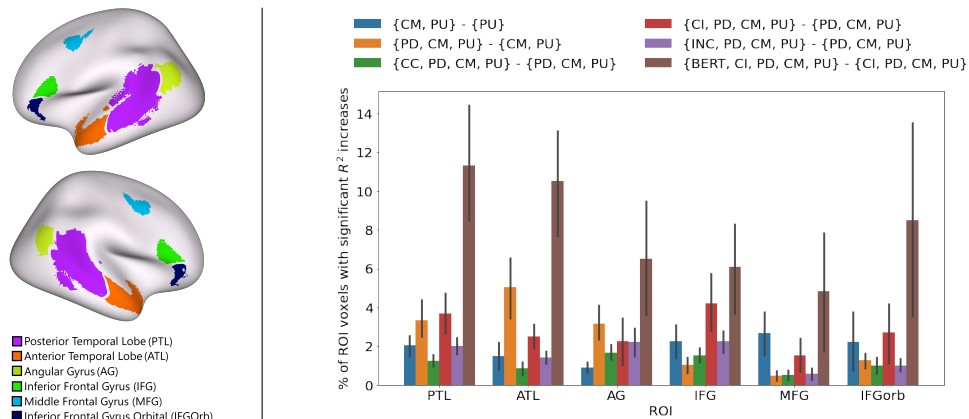

Figure 4: Region of Interest (ROI) analysis of the prediction performance. [Left] Language system ROIs by [26] from [44]. [Right] For each model, the percentage of ROI voxels in which we see significant increases in prediction performance. Each bar represents the average percentage across subjects and the error bars show the standard error across subjects. We use the same abbreviations as in fig. 3 and see the same trends across ROIs.

| Feature | Level 2 | Level 3 | Level 4 | Level 5 | Level 6 | Level 7 | Level 8 | Level 9 |
|---|---|---|---|---|---|---|---|---|
| Most Popular Label % | 51 | 38.76 | 54.42 | 64.05 | 73.44 | 78.25 | 82.38 | 85.82 |
| All Complexity Metrics | 49.84 | 45.29* | 59.66* | 67.99* | 77.3* | 82.13* | 86.34* | 89.74* |
| POS and DEP tags | **92.23*** | **71.27*** | **67.06*** | **70*** | **77.51*** | 82.09* | 86.26* | 89.59* |
| ConTreGE Comp | 66.76* | 51.01* | 59.52* | 67.95* | 77.28* | 82.08* | 86.26* | 89.68* |
| ConTreGE Incomp | 52.91 | 45.42* | 57.37* | 67.29 | 76.87* | **82.3*** | **86.51*** | **90.39*** |
| InConTreGE | 52.46 | 45.59* | 57.61* | 66.75 | 76.13 | 81.08 | 85.27* | 89.1* |
| BERT Embeddings | 52.18 | 45.73* | 58.62* | 66.79 | 75.68 | 80.31 | 84.72 | 88.72 |

Table 1: 10-fold cross validation accuracies in predicting the ancestors of a word. * denotes accuracies significantly above chance (tested using Wilcoxon signed-rank test, $p \leq 0.01$). POS and DEP tags best predict lower level ancestors while ConTreGE Incomp vectors best predict higher level ones.

**Syntactic structure embeddings capture additional variance in the brain activity even after controlling for complexity metrics**  From fig. 3(b), it can be seen that the information provided by the complexity metrics is predictive of brain activity to a certain degree even after controlling for punctuation. Fig. 3(c)-(f) show that when the features that encode syntactic structure are added to the feature groups, they explain even more variance, showing that fMRI can indeed reveal syntactic representations. These results are made even clearer by fig. 4. The $R^2$ scores in many voxels increase significantly (above what is predicted by the complexity metrics) after including POS and DEP tags and ConTreGE Incomp. ConTreGE Comp is not as predictive as ConTreGE Incomp. Additionally, InConTreGE is not as predictive as ConTreGE Incomp, suggesting that the top-down parser might be generating partial parses that are not reflective of brain representations.

**Complex syntactic information is distributed in the brain**  In this section, we analyze the information in ConTreGE Incomp to interpret its brain prediction performance. We estimate how much of the constituency tree is captured by each feature by using it to predict the level N ancestor of a word (in its constituency tree). We vary N from 2 to 9 and train a logistic regression model for each N. Since POS tags are the level 1 ancestors of words, we start the analysis at N=2. As there are many phrase labels, we group them into 7 buckets - noun phrases, verb phrases, adverb phrases, adjective phrases, prepositional phrases, clauses and other miscellaneous labels. If a word's depth in its tree is less than N, the root is considered its level N ancestor.

Table 1 shows the results of this analysis. We use the trees generated by the Kitaev and Klein [32] parser. Given the skewed label distribution, the optimal strategy for a predictor that takes random noise as input is to always output the majority class ancestor at that level. Chance performance is thus equal to the frequency of the majority label. POS and DEP tags are predictive of labels at all levels and produce the highest accuracies for lower levels. The InConTreGE vectors are not as predictive as ConTreGE Incomp or ConTreGE Comp, hinting that the top-down parser might not be very accurate. ConTreGE Incomp is the best predictor of higher level ancestors but ConTreGE Comp is better than ConTreGE Incomp at predicting lower level ancestors. This may be because graph embeddings of a tree tend to capture more of the information near the tree's root (a random walk through a somewhat balanced tree is likely to contain more occurrences of nodes near the root and random walks are encoded in the subgraph embedding generation process). ConTreGE Comp vectors, created from shallow complete trees, likely over-represent lower level ancestors while ConTreGE Incomp vectors, created from relatively deeper trees, likely over-represent higher level ancestors. Given that ConTreGE Incomp contains information about the higher level ancestors of a word, this suggests that we are able to pick up on the distributed brain representations of complex hierarchical syntactic information such as phrase and clause structure, using this feature.

The ConTreGE Incomp vectors also encode future syntactic constituents. Thus, it is possible that this information is helpful in predicting the current brain activity. Although this result supports the hypothesis that the brain predicts future syntactic structure, we cannot conclusively prove it. This is because certain other characteristics of the feature space such as its smoothness might also make it more predictive. We leave it to future work to explore this hypothesis in more depth.

**Syntax and semantics are processed in a distributed way in overlapping regions across the language system**  We first empirically show that our ConTreGE vectors do not encode a significant amount of semantics. We run a simple analysis similar to the one used by Caucheteux et al. [47]. Like in Caucheteux et al. [47], we try to predict a GloVe-based semantic vector [48] (extracted using spaCy [37]) using the BERT embeddings and the ConTreGE vectors. These predictions were performed using a simple ridge regression model whose regularization parameters were chosen based on leave-one-out cross-validation $R^2$ scores (performed using sklearn's RidgeCV module). In order

to get the final $R^2$ scores, we perform 10-fold cross-validation with a separate RidgeCV model being fit for each train-test split. We find that the BERT embeddings achieve average $R^2$ scores of 0.185 (the $R^2$ scores are first averaged across all 300 dimensions of the GloVe-based semantic vectors and then across all 10 cross-validation splits). However, the graph embeddings achieve significantly lower scores with ConTreGE Comp, ConTreGE Incomp and InConTreGE achieving average $R^2$ scores of 0.052, 0.020 and 0.021 respectively. These scores are markedly lower than the ones obtained using BERT, illustrating that the graph embeddings contain very low amounts of semantic information. On the other hand, our syntactic information analysis (Table 1) shows that these embeddings capture a lot of syntactic information. All of our other syntactic feature spaces are purely syntactic by definition.

Since the synactic feature spaces encode a lot of syntactic information, it is reasonable to assume that any additional variance predicted by the BERT embeddings after controlling for the syntactic feature spaces, is at least partially due to their semantic information. From Figures 3 and 4, we see that the BERT embeddings explain additional variance in most of the regions in the language system and these regions overlap with the regions that are predicted by the syntactic feature spaces. As we have already established that the syntactic feature spaces contain a small amount of semantics, we can thus conclude that syntactic and semantic information are processed in a distributed fashion across the language network and that there are no regions that are selective for syntax.

## 5   Discussion and Related Work

**Syntactic representations**   Multiple complexity metrics have been proposed. Hale et al. [9] use parser action count (the number of possible actions a parser can take at each word) to model EEG data. They find that it is predictive of the P600, a positive peak in the brain's electrical activity occurring around 600ms after word onset. The P600 is hypothesized to be driven by syntactic processing (to resolve incongruencies), and the results of Hale et al. [9] align with this hypothesis. Brennan et al. [22] derive the surprisal and node count metrics using nine different models (metrics are computed only if the model allows their computation, e.g. Node Count cannot be computed using Markov models). Each model captures syntactic information at one of three levels of abstraction - most basic word-to-word dependencies (Markov models), hierarchy of phrases (context-free PSG) and the most abstract level which captures long-distance dependencies (Minimalist Grammars [49, 50]). They observe that the metrics computed using the hierarchical grammars are predictive of fMRI signals from the left ATL and PTL which were collected during natural listening. The metrics derived from the Markov models predict the same regions along with the IFG.

Apart from these studies, many others [10, 11, 22, 24, 25, 51] use complexity metrics to study syntactic processing during natural reading or listening. However, a few studies do explicitly encode syntactic structure: Wehbe et al. [17] find that POS and DEP tags are the most predictive out of a set of word, sentence and discourse-level features. Concurrent with our work, Caucheteux et al. [47] also try to build syntactic representations using GPT-2 [52] embeddings and obtain results similar to ours. Complementing popular approaches that are dependent on complexity metrics, we built upon the work of Wehbe et al. [17] by developing a novel graph embeddings-based approach to explicitly capture the syntactic information provided by constituency trees. Our results show that these explicit features have more syntactic information than complexity metrics and this information is able to predict additional variance in the brain activity. Given these results, we believe that future work in this area should supplement complexity metrics with features that explicitly encode syntactic structure.

**Syntax in the brain**   Traditionally, studies have associated a small number of brain regions, usually in the left hemisphere, with syntactic processing. These include parts of the IFG, ATL and PTL [2, 3, 53, 54]. However, some works point to syntactic processing being distributed across the language system. Blank et al. [4] show that there are significant differences in the activities of most regions of the system when phrases that are harder to parse are read compared to when easier phrases are read. Using a comprehensive set of experiments, Fedorenko et al. [6] show that the entire fronto-temporal language network responds to syntax. Wehbe et al. [17] use POS and DEP tags to arrive at similar conclusions.

Previous work generally did not use naturalistic stimuli to study syntax. Instead, subjects are usually presented with sentences or even short phrases that have subtle syntactic variations or violations. Regions with activity well correlated with the presentation of such variations/violations are thought to process syntax [3]. Observations from such studies have limited scope since these variations often

cannot be representative of the wide range of variations seen in natural language. This is possibly why such studies report specific regions: it might be that the reported region is particularly sensitive to the exact conditions used. By using one type of stimulus which evokes only one aspect of syntactic processing, syntax might appear more localized than it really is. Our results support the hypothesis that it is instead processed in a distributed fashion across the language system. We believe that our results have a wider applicability since we use naturalistic stimuli and we leave for future work the study of whether different syntactic computations are delegated to different regions.

Some studies have also doubted the importance of syntactic composition for the brain. Pylkkänen [55] proposes that there is no conclusive evidence to indicate that the brain puts a lot of weight on syntactic composition, and that even though studies (some with complexity metrics) have associated certain regions like the left ATL with syntactic processing, numerous studies have later shown that the left ATL might instead be involved in a more conceptually driven process. Gauthier and Levy [56] showed that BERT embeddings which were fine-tuned on tasks that removed dependency tree-based syntactic information were more reflective of brain activity than those which contained this information. In contrast, our work uses purely syntactic embeddings to show that we can indeed significantly predict many regions of the language system. We attribute these differences in conclusions to our naturalistic stimuli and word-by-word evolving representations of syntax. Pylkkänen [55]'s conclusions are mostly based on studies that present a phrase with just two words (like "red boat"). Gauthier and Levy [56] use data averaged over entire sentences instead of modeling word-by-word comprehension. Since the syntactic structure of a sentence evolves with every word that is read, this approach is not necessarily adept at capturing such information.

Furthermore, our analysis of the information contained in various features highlighted that our ConTreGE Incomp vectors are good at encoding complex phrase or clause-level syntactic information whereas POS and DEP tags are good at encoding local word-level information. Several regions of the brain's language system were predicted by ConTreGE Incomp, hinting that these areas encode complex syntactic information. Another potentially interesting observation is that including ConTreGE Incomp increases prediction performance in the PTL and IFG by more than when we include POS and DEP tags (Figure 4) but not for the ATL and the AG. These observations very loosely support the theory by Matchin and Hickok [54], which stipulates that parts of the PTL are involved in hierarchical lexical-syntactic structure building, the ATL is a knowledge store of entities and the AG is a store of thematic relations between entities. This is because ConTreGE Incomp encodes hierarchical syntactic information and word-level POS and DEP tags are very indicative of the presence of various entities (various types of nouns) and the thematic relations between entities (verbs associated with noun pairs). This hypothesis should be tested more formally in future work.

**Syntactic vs. semantic processing in the brain**    Finally, our results support the theory that syntax processing is distributed throughout the language network in regions that also process semantics. This theory is supported by other studies [5, 4, 6]. However, Friederici et al. [53] among others argue that syntax and semantics are processed in specific and distinct regions by localizing the effects of semantic and syntactic violations.

**Limitations of our approach**    Our first step towards studying the *representations* of syntax in the brain still has a few limitations. First, we cannot fully encode the information present in dependency trees using our graph embeddings-based approach. This is because the nodes of these trees are the words themselves and the DEP tags identify the edges between them. Our method cannot encode edge labels, it only encodes node labels. Second, we only had access to data from 9 subjects who were asked to read English text. To determine the generality of our results, data from other experimental setups and languages will need to be analyzed using our features. Like other neural embeddings, our ConTreGE embeddings also trade off interpretability for expressivity. Although we try to deconstruct them, they are much harder to understand when compared to the simple complexity metrics or POS and DEP tags. We also try to limit the amount of semantics encoded in the embeddings. However, because of our usage of natural text, there are bound to remain some correlations between semantics and syntax that cannot be removed due to the nature of human languages. Finally, the incremental top-down parser we use does not produce very good parses when compared to the more modern neural parser, limiting the concreteness of the conclusions about the InConTreGE results.

## Acknowledgments and Disclosure of Funding

LW and AJR were supported by startup funds at Carnegie Mellon University. The authors would like to thank Jennifer Williams, Mariya Toneva and Srinivas Ravishankar for useful feedback on the manuscript.

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
