# OpenReview forum: "Can fMRI reveal the representation of syntactic structure in the brain?"
_NeurIPS.cc/2021/Conference — NeurIPS 2021 Poster_

### Official Review · Reviewer_KgNW · 2021-07-11

**Rating:** 6
**Confidence:** 3

**Summary:**

* The authors conduct a brain encoding study testing whether fine-grained information about the content of incremental syntactic representations contribute to predictions of neural activity, over and above measures of syntactic complexity.
* They design custom subgraph embedding representations to operationalize the idea of "syntactic representation." They find that these subgraph embeddings capture additional variance in brain activity above what is predicted by complexity metrics, and that this improvement in prediction is distributed throughout the language system.

**Limitations And Societal Impact:**

Yes.

**Main Review:**

The authors confront the important question of the representation of syntactic content in brain activity during language processing. I appreciate their creative choice of graph embeddings as a more specific, but still somewhat theory-free, proxy for mental representations of syntax. The paper could use improvements both in interpretation of results and in organization/clarity. But I believe the authors are making a good contribution to an important conversation. I mark the paper as marginally below acceptance threshold, but would change my review if the authors can (minimally) adequately address the theoretical concerns in the "Quality" section below.

## Quality

### Are ConTreGE embeddings distinctly syntactic?

There seem to be two distinct claims that are important to the authors, and which they take their results to support:

1. Syntactic representational content at a level deeper than complexity measures is present in recordings of brain activity. \*ConTreGE\* embeddings contain some of this representational content, and thus succeed at brain encoding over and above the predictions of complexity measures.
2. \*ConTreGE\* embeddings are distinctly syntactic, and so there exist brain regions selective for syntax iff there are regions well predicted by \*ConTreGE\* embeddings but not by BERT embeddings.

My hunch is that while \*ConTreGE\* embeddings do contain meaningful syntactic content, they also contain (correlates of) semantic content. This is because constituency trees still retain information at the syntax--semantics boundary. For example, in the PTB syntactic representation it is possible to read off semantic features such as number, aspect (from correlated tense features), and modality (from the presence/absence of auxiliaries).

Under this hunch, \#1 is valid but \#2 is not. Overlap in well-predicted regions between the \*ConTreGE\* embeddings and BERT embeddings (after controlling for ConTreGE success) may be due to the fact that the \*ConTreGE\* embeddings are not distinctly syntactic.

I would like the authors to address this question more directly. They do include (in lines 285--308) a test of the syntactic representational content of the derived ConTreGE embeddings, but this is of course not the same as demonstrating that the embeddings are distinctly syntactic. They might include more tests of representational content (or lack thereof) in the ConTreGE embeddings (e.g. visualizations of principal components of embedding space, decoding experiments, etc.), or post-hoc qualitative or quantitative evaluations of the differences between ConTreGE and BERT embeddings in real linguistic contexts.

As I understand it, this premise that ConTreGE embeddings are distinctly syntactic only matters for the inference about the overlap between syntactic and semantic content in neural representations. If there are other claims depending on this premise, I think they should also be reevaluated.

### Comparing the representations

I have several small qualms with interpretation of the differences between brain encoding with the different representations:

1. The relative success of the Incomp representation is taken as evidence that the brain is modeling future syntactic information (lines 281--283). But Incomp can contain more context than Comp in both directions. This is evident in the Comp--Incomp contrast in Figure 1B/1C: the Incomp representation carries information about content appearing before the largest completed subtree as well as after. Because of this, I don't think you can use the relative success of Incomp to argue for representations of future syntactic content.
2. Why do we believe that the Incomp representation should not represent uncertainty about the structure governing past content? (Why does Fig. 1C assume that "believed" is VBD? How is VBD chosen here over the alternative VBN as Fig. 1E makes apparent?)

## Clarity

The paper could benefit from some reorganization. The introduction is extremely heavy at the moment (a full two pages). I would recommend the authors compress this section and use some of the saved space to clearly outline the embedding methods and run intrinsic analyses / visualizations on these embeddings before moving on to brain encoding work.

## Other minor comments

1. I'm not a neuroscientist and can't address the validity of the encoding method used. I was slightly surprised to see the authors chose to learn a mapping from all the representations triggered in a TR, rather than using a convolution based on a hemodynamic response function.
2. Fig. 1e --- How is this a valid parse? If this is supposed to be a relative clause construction, the constituency structure is incorrect; the VBN should be a child of the NP. Also, an RC is implausible here with "Harry" -- you'd want a nonlexical NP such as "the person."

**Time Spent Reviewing:**

2

---

> ### Author Response · Authors · 2021-08-10
> **Response to Reviewer KgNW**
>
> We thank the reviewer for their comments. This reviewer and reviewer djyf both commented on the assumptions behind the ConTreGE and BERT embeddings, and we will clarify them here again. We do not say that the BERT embeddings do not contain syntax. We make an assumption however that what syntax they do contain is in big part contained in the ConTreGE embeddings and the other syntactic features, and that the additional variance predicted by BERT *after controlling for all our syntactic features* is due to the semantic information in BERT. Since there are no regions that are predicted by any of the syntax embeddings and not by BERT, this suggests that there are no regions that are predicted by syntax but not by semantics.
>
> Even if it was the case that ConTreGE has semantic information due to natural correlations, it is still the case that there was no additional area that was predicted by ConTreGE or the other syntax features and not by BERT. This is the basis for our conclusion that syntax regions are not selective for syntax. In order to conclude that a region is selective for syntax, the prediction performance in the region must not increase when BERT is added.
>
> More about the nature of the information in ConTreGE: We build the ConTreGE vectors out of constituency trees that do not contain the word identity as leaves. Instead, words are replaced by numbers from 1 to n (total number of words in the text presented). This is done to minimize the amount of semantic information. Parse trees are some of the most fundamental sources of syntactic information. We agree with the reviewer that there may remain some correlations between these trees and semantic information because of our usage of natural text. However, such correlation between semantics and syntax is an unremovable part of natural language. Since the brain processes such natural material every day, such correlations could very well be integrated into how the brain processes these types of information (i.e. the brain could be learning these correlations and processing syntax or semantic information using them).
>
> Finally, in order to empirically show that our ConTreGE vectors do not encode a significant amount of semantics, we run a simple analysis similar to the one used by Caucheteux et al. (2021). Like in Caucheteux et al. (2021), we try to predict a GloVe-based semantic vector (extracted using spaCy) using the BERT embeddings and the ConTreGE vectors. These predictions were performed using a simple ridge regression model whose regularization parameters were chosen based on leave-one-out cross validation $R^2$ scores (performed using sklearn’s RidgeCV module). In order to get the final $R^2$ scores, we perform 10-fold cross validation with a separate RidgeCV model being fit for each train-test split. We find that the BERT embeddings achieve average $R^2$ scores of 0.185 (the $R^2$ scores are first averaged across all 300 dimensions of the GloVe-based semantic vectors and then across all 10 cross-validation splits). However, the graph embeddings achieve significantly lower $R^2$ scores with ConTreGE Comp, ConTreGE Incomp and InConTreGE achieving average $R^2$ scores of 0.052, 0.020 and 0.021 respectively. These scores are markedly lower than the ones obtained using BERT, illustrating that the graph embeddings contain very low amounts of semantic information. On the other hand, our syntactic information analysis (Table 1), shows that these embeddings do indeed capture a lot of syntactic information. We will make space in the paper to discuss the comments above and to explain this analysis.
>
>
> Regarding Comparing the representations:
>
> - While it is true that the Incomp vectors encode a lot more context than Comp in both directions, note that the higher level ancestors of a word are often unknown until many future words are read (especially in long sentences). Thus, even the knowledge of these ancestors (which may be on the left of a word) constitutes future syntactic information. Furthermore, because of the temporal resolution of fMRI and the delay in the hemodynamic response, we need to use the embeddings of about 16 consecutive words while predicting the fMRI signal (described in lines 229-235). This usage of consecutive embeddings should further bridge the gap in the amount of left context encoded in the Comp and Incomp vectors. These reasons make us believe that the relative success of the Incomp vectors hints at the brain modeling future syntactic information.
>
> - While constructing the Incomp vectors, we only use the most probable full sentence parse output by the self-attentive encoder-based parser [25]. We have created this anonymized document that shows more examples of complete and incomplete trees for a complex sentence - https://drive.google.com/file/d/1TZgzJAtjNoJyeAfyDyGHJeZbYqFDknFP/view?usp=sharing. We hope this document can help in understanding the generation process. We do not consider the uncertainty involved in constructing such a parse while constructing these vectors as the InConTreGE vectors are used for exactly this purpose. The VBD tag is the tag assigned to the word in the most probable parse. The VBN tag referenced in Fig 1E is from one of the partial parses output by the incremental top-down parser [27]. These partial parses are encoded in the InConTreGE vectors so as to capture uncertainty.
>
> We will follow the reviewer’s recommendation to reorganize the paper (reviewer djyf also made a similar suggestion).
>
> Other minor comments:
> - The delay approach used in the paper is actually common in encoding models and is often preferred to using a fixed canonical response. The assumption that the response is fixed and the same everywhere in the brain is not correct. The delay approach instead allows us to implicitly learn the hemodynamic response in each voxel. (Kay et al 2008 Human Brain Mapping, Nunez-Elizalde et al 2019 Neuroimage).
> - This was just one of the partial parses output by the incremental top-down parser [27] for this sentence. We did not check it for its validity. We will be sure to include a better example in the final version of the paper.
>
> References
>
> Caucheteux, Charlotte, Alexandre Gramfort, and Jean-Remi King. "Disentangling syntax and semantics in the brain with deep networks." International Conference on Machine Learning. PMLR, 2021.

---

> > ### Comment · Reviewer_KgNW · 2021-08-25
> > **Response to response**
> >
> > ## Summary of response
> >
> > - I don't think the authors have adequately addressed my (and djyf's) principal concern about the possible traces of semantic content in ConTreGE embeddings, and the effects those traces would have on their conclusions about the overlap between syntax and semantics.
> > - Regarding "Comparing the representations," I think these claims deserve an entirely separate paper with experiments specifically designed to test questions related to predictive coding. I don't think it's scientifically responsible to include claims about predictive coding (or "hints" of predictive coding!) as a minor offshoot of the larger experiment.
> >
> > Overall, I think the motivation of this paper is admirable and the work is very promising. I just think that more careful attention is needed on the theoretical ideas and the empirical tests of those ideas. I am keeping my score a 5 for this reason.
> >
> > ## Syntactic and semantic representational content
> >
> > > We do not say that the BERT embeddings do not contain syntax.
> >
> > Just to be clear, this was not at issue -- speaking for myself, not for djyf.
> >
> > > We make an assumption however that what syntax they do contain is in big part contained in the ConTreGE embeddings and the other syntactic features, and that the additional variance predicted by BERT after controlling for all our syntactic features is due to the semantic information in BERT.
> >
> > This is indeed a necessary assumption, but also not the one I took issue with. I call this the "exhaustive" assumption below, but am really more concerned with the "exclusive" assumption. See the next paragraph.
> >
> > > In order to conclude that a region is selective for syntax, the prediction performance in the region must not increase when BERT is added.
> >
> > This is only convincing if you grant the assumptions that ConTreGE embeddings are 1) exhaustively (relative to BERT) and 2) exclusively syntactic. I'm happy to grant #1 since I'd wager *a priori* it's less of an issue. But if #2 is false, your result can emerge just because the regions in which BERT fails to improve predictions are well-predicted due to the (correlates of) semantic content within ConTreGE.
> >
> > > We agree with the reviewer that there may remain some correlations between these trees and semantic information because of our usage of natural text. However, such correlation between semantics and syntax is an unremovable part of natural language. Since the brain processes such natural material every day, such correlations could very well be integrated into how the brain processes these types of information (i.e. the brain could be learning these correlations and processing syntax or semantic information using them).
> >
> > Your answer seems to beg the critical question. If you grant that syntactic and semantic content are correlated in natural language, and that the brain ought to exploit such dependencies in its processing, why is it of interest to test whether syntactic and semantic representations are distinctly represented in the brain?
> >
> > It’s logically possible that the two representations have systematic correlations with one another, but that they also be mechanistically / spatiotemporally dissociated in the mind/brain. But a test of the latter claim needs to rigorously 1) identify and 2) control for these systematic correlations.
> >
> > > These scores are markedly lower than the ones obtained using BERT, illustrating that the graph embeddings contain very low amounts of semantic information.
> >
> > Frankly, given the weight of the claim you are trying to make (about patterns in linguistic representation across the whole language processing hierarchy), I don't think this evaluation does enough to support the premise of syntactic exclusivity. I would be strongly in favor of linguistically informed tests, broad- or narrow-coverage, that address some of the a priori concerns about correlated representational content that I and djyf mentioned. Concretely -- with what accuracy can you decode the number of the sentence's subject? different types of modal meanings? etc.

---

> > > ### Author Response · Authors · 2021-08-26
> > > **Response**
> > >
> > > We thank the reviewer for their comments. We agree with the reviewer about the exhaustive and exclusive assumptions needed to arrive at our conclusions in the general case. However, because of the patterns presented in our results, only the exhaustive assumption is required to conclude that there are no regions that are selective for syntax. This is because there are no broad regions that are well-predicted by the ConTreGE vectors but not by BERT. This means that the argument presented in the below sentence does not change our conclusion if the exhaustive assumption is granted as there are no broad regions where BERT fails to improve predictions:
> > >
> > > > But if #2 is false, your result can emerge just because the regions in which BERT fails to improve predictions are well-predicted due to the (correlates of) semantic content within ConTreGE.
> > >
> > > In other words, we present a proof by contrapositive. The first statement is: if there is a broad region that is selective for syntax, the prediction performance in that region would not improve by adding more semantic information. We show that adding semantic information (contained in the BERT embeddings) improves prediction performance everywhere. As we do not see a drop in performance in any region, the original premise (that there is a region that is selective for syntax) is suggested to be false.
> > >
> > > Whether the ConTreGE vectors encode minute amounts of semantics on top of syntax doesn’t affect our logic. Consider a possible area A that only processes syntax. If this area is predicted by ConTreGE, and then its prediction performance improves further using BERT (after controlling for ConTreGE), there is a contradiction, and it cannot be selective for syntax. Whether ConTreGE is purely syntactic or whether it encodes some semantics doesn’t interfere with our argument (as long as it encodes a sufficient amount of syntax).
> > >
> > > Another important question the reviewer asked is “If you grant that syntactic and semantic content are correlated in natural language, and that the brain ought to exploit such dependencies in its processing, why is it of interest to test whether syntactic and semantic representations are distinctly represented in the brain?” In fact, this question is of utmost interest and is one of the core motivations of this paper. This is a crucial question in the neurobiology of language field. We invite the reviewer to check the results of Friederici et al. (2003) [47] and to contrast them with our results and those of [4, 5, 6]. Fedorenko et al. (2020) [6] reviews prior literature on this question. A large part of the neurobiology of language community considers that some areas are selective for syntax. However, when you use naturalistic text and an encoding model, this selectivity disappears. We feel that this finding is important for the field and should be reported.
> > >
> > > We ask the reviewer to consider the syntax vs. semantics argument from this angle, and as answering a neurobiology of language question, not a psycholinguistics one. One of the contributions of this paper is to weigh in on this existing debate using evidence from an encoding model-based setup. We will clarify the logic of our argument in the main text.
> > >
> > > Regarding the predictive coding – we mainly present these findings to showcase the nature of questions that can be answered using our encoding method. We do not make very strong statements regarding this question as we are aware of its complexity. However, to the best of our knowledge, there are no other encoding model-based studies which have tried answering this question. Thus, we felt that presenting this question and our findings might be helpful to start a discussion on this topic. As the reviewer suggests, this question deserves a separate paper, and we leave it to future work to perform a more rigorous analysis.

---

> > > > ### Author Response · Authors · 2021-09-01
> > > > **Have the reviewer's concerns been addressed?**
> > > >
> > > > Since the discussion period ends soon, we wanted to know if the reviewer's concerns have been addressed by our most recent comments. Please do let us know if you have any additional questions. Thank you!

---

> > > > ### Comment · Reviewer_KgNW · 2021-09-10
> > > > **Response response response response response**
> > > >
> > > > Hi authors, thanks for kindly clarifying here. My apologies for the last-minute response. My memory of the results seems to have been muddled in my last response, as evidenced by the sentence
> > > >
> > > > > your result can emerge just because the regions in which BERT fails to improve predictions are well-predicted due to the (correlates of) semantic content within ConTreGE
> > > >
> > > > ... which isn't true for any broad region, as you reminded me above. Your proof-by-contradiction logic makes sense to me. (I hope you will revise lines 108--117, which I believe is where I got the idea that the exclusivity assumption was crucial to your argument.)
> > > >
> > > > ---
> > > >
> > > > This was my major objection and I now understand it wasn't well-founded given the results. As such I'll revise my rating to a 6.

---

### Official Review · Reviewer_zeMS · 2021-07-12

**Rating:** 7
**Confidence:** 5

**Summary:**

In this paper, the authors propose new representational spaces that capture syntactical information in natural language above and beyond complexity-based metrics that have been typically used in language neuroscience. First, they build representations based on the constituency tree of a sentence as every word is processed by the parser. Then, they use a subgraph embedding algorithm to convert the subtree into a 15-D representation. To test several hypothesis about the type of syntactic information being encoded, the authors look at 3 different types of subtrees- the largest subtree completed when the current word is processes, the incomplete subtrees that can explain the sequence seen thus far under PSG production rules and the set of complete parses produced by a probabilistic model that are weighted by their probabilities. The 3 subspaces are compared against 4 complexity/load-based metrics and additionally, a semantic feature space derived from BERT. They go on to build encoding models for the different feature spaces and compare them through variance partitioning. Overall, the paper finds that the 3 graph-based spaces explain additional variance across the cortex as compared to traditional complexity-based features. Further, this effect is not localized to any region. Instead, it seems to be distributed across a network that is largely explained by semantic features, suggesting that syntax and semantic share neural substrates and there exists no isolated region for syntactic processing.

The study uses publicly released fMRI data from 9 subjects reading a chapter from Harry Potter .

**Limitations And Societal Impact:**

The paper contains information on the limitations of the approach specifically but not broad societal impacts of encoding models and neuroimaging techniques.

**Main Review:**

Overall, this paper was well-written and easy to understand. While it would help to clarify how the 3 important ConTreGE features are computed for different constituents/nodes, I thought the idea of breaking down syntactic processing into 1) current subtree 2) anticipated subtree 3) all possible subtrees was thoughtful. Although the implications on language neuroscience are not novel (other work has shown or hinted that syntax and semantics likely have shared neural processing), to the best of my knowledge this is the first work demonstrating it in an encoding model setup with feature spaces that go beyond traditional effort-based metrics.

Some clarifications & suggestions:
- Re ConTreGE vectors:
    - How many words in a sentence have different representations than POS Tags? For instance, all specifiers/heads would essentially have the same information as their POSTags but a complement or modifier might not right? It is unclear to me what the authors mean by
> the largest subtree that is completed
For instance, why isn’t VP a valid subtree for _believed_ assuming that this verb has no complement? I understand that this is a secondary concern given that the representations are subgraph embeddings and thus, richer than the POSTags in many instances. But it is unclear to me how many times this is actually true and what are the rules that govern subtree selection. (My confusion was reinforced by the results of PD vs. CC in Fig. 3 & 4 + given that the random walk is still on nodes and not edges)
- Re ConTreGE Incomp vectors:
    - The example provided in Fig. 1C is used as a feature for _believed_ or _it_? I presumed _believed_ based on the hypothesis that the brain is already predicting future structure when it hears the word _believed_. It is unclear to me what this would look like for an already complete constituent, like _Harry_. Would this be any different from ConTReGE Comp? I agree that the idea of predictive coding (or so to speak) is interesting in terms of syntax, but do we really have the temporal resolution for this? Given that the word-space is downsampled and several of the features are very similar to the complete trees, I would like to know what the authors think.
- Re Incremental ConTreGE: How does this compare to say, using the softmax layer of a POSTagger? [I also think it would be more demonstrative to add examples of sub-trees where the structure itself is ambiguous: Maybe like a garden path sentence?]
- How do you deal with bidirectionally in BERT, given that the fMRI responses are generated causally?
- I had a hard time interpreting Fig. 3:
    - If the values are discrete from 0-5 , why not use a discrete & distinct colormap like viridis? It is not obvious to me what anything but dark green/white means.
    - Since the subjects were coregistered to a common space, I think this figure might be underselling performance. For instance, its possible that one subject has lower prediction overall due to SNR etc.. To this end, wouldn’t plotting just the number of subjects underestimate the unique variance explained by the new feature space?
    - What are the $R^2$ values? It seems that adding the ConTreGE features doesn’t improve performance substantially but this could be an incorrect reading of the figure. In addition to Fig. S5, I would appreciate a 3D model showing the unique variance explained for ever feature space that is added (and not the concatenation). Also, what is the significance threshold for the correlation values plotted in Fig. S5?
    - Missing reference that this is essentially variance partitioning [deHeer et al., 2017]
    - It seems strange to me that not enough voxels & subjects in the frontal cortex benefit from BERT when several studies (including the ones cited in this paper) have shown robust responses to semantic features in that part of the brain (especially given that none of the syntax models predict well there) Although, Fig. S5(k) suggests high correlation.
- My comments on ease of interpretation also apply to Fig. 4 wherein it would be helpful to show the percentage of significant voxels under each model as an individual bar. (I understand that it might be too crowded; but maybe stacked bars?)
    - I find it interesting that comparable gains are found between BERT and some of the syntactical feature spaces in Broca’s area (contained in IFG) and sPMv (contained in MFG).
- How are the word features downsampled? Would be useful to add this to the paragraph starting in Line 229.
- For the _bootstrap_ test in lines 251-255: I am confused if the TR blocks were sampled with replacement like in a bootstrapping procedure or were randomly permuted like in a permutation test.
- What does the sliding window refer to in line 188?
- It would help to be more consistent in citing work that uses encoding models for fMRI. For instance, several publications that use encoding models for fMRI (JR King’s group, Jelle Zuidema’s group, Uri Hasson’s group, Alex Huth’s group) have been elided and there is inconsistency in what publications from the references are used at specific points (lines 63-64).
- Missing reference: Bhattasli et al., 2018 is closely related to _parser action count_ in Hale et al., 2018.

I am fairly certain who the authors are from having seen a previous version of the paper + a talk but the authors did not violate anonymity in any way.

**Time Spent Reviewing:**

5

---

> ### Author Response · Authors · 2021-08-10
> **Response to Reviewer zeMS**
>
> We thank the reviewer for their comments. Inline answers below.
>
> - *it would help to clarify how the 3 important ConTreGE features are computed for different constituents/nodes:*
>     - They are computed uniformly irrespective of the specific constituent/node.
>
> - *Re ConTreGe: How many words in a sentence have different representations than POS Tags? … what the authors mean by the largest subtree that is completed:*
>     - Roughly 2440 out of the 5176 words presented  (~47%) have completed trees that consist of higher level nodes.
>     - A subtree is considered complete when all of its leaves are terminals. The largest subtree completed by a word refers to the subtree with the largest height that also satisfies the following conditions:
>         1. The given word must be one of its leaves
>         2. All of its leaves must only contain words that have been seen till then.
>
> -  *Re: ConTreGe Incomp: The example provided in Fig. 1C is used as a feature for believed or it?*
>     - Yes, all of these subtrees are for "believed". We will clarify this in the final version.
>     - We have created this anonymized document that shows more examples of complete and incomplete trees for a complex sentence - https://drive.google.com/file/d/1TZgzJAtjNoJyeAfyDyGHJeZbYqFDknFP/view?usp=sharing. We hope this document can help in understanding the generation process.
>     - Regarding the temporal resolution - consider the incomplete subtrees for the words “Harry” and “Draco” shown in the above document. These two words are separated by more than 5 TRs (10 seconds, enough to resolve differences in the hemodynamic response) and have very different incomplete subtrees. These incomplete subtrees are also very different from their complete counterparts. Thus, these longer sentences enable us to study predictive coding. Out of the 437 sentences in the presented text, 164 sentences have more than 15 tokens.
>
> -  *Re: Incremental Contrege: How does this compare to say, using the softmax layer of a POSTagger?*
>     - Using the softmax layer of a POS tagger would only allow us to capture ambiguities in the lexical syntactic structure. By using the partial parses output by the incremental top-down parser [27], we can capture ambiguities in both the lexical and compositional syntax of a sentence, making it more powerful in theory.
>     - We just used two short partial parses output by the top-down parser due to space constraints, we will be sure to include better examples in the final version of the paper.
> - *How do you deal with bidirectionally in BERT, given that the fMRI responses are generated causally?*
>     - We only generate embeddings using the words of the sentence that have been seen. We do not input the words that have not been seen (l209-210).
> - *Figure 3*:
>     - We will use a more distinct colormap like viridis
>     - We agree that the conversion to standard space reduces the importance of findings that have more spatial variance. One solution could be to report the average prediction performance which is still smoothed out over such regions but doesn’t suffer from the somewhat arbitrary threshold for significance. These maps would correspond to the difference of the maps in supplementary figure 5.
>     - The raw $R^2$ scores in supplementary figure 5 show the prediction performance obtained using the hierarchical feature groups (before the subtraction is performed for significance testing). They are not thresholded.
>     - Good point, we will include the de Heer et al. variance partitioning reference
>     - The apparently low number of significant increases might be because the increase is not consistent across subjects. This might be due to the limited signal to noise ratio afforded by this dataset. It is definitely the case that some of the subjects show a significant increase for BERT in many frontal voxels.
> - *Fig 4 percentage of significant voxels*
>     - We can definitely add a figure or a set of figures with these bars.
> - *How are features downsampled*
>     - The features are summed across the words for each TR (2 seconds = 4 words). The features are then delayed by 1, 2, 3 and 4 TRs and concatenated. This is mentioned in lines 229-235. We can add more clarifying information if needed.
> - *bootstrap test in lines 251-255*
>     - The TR blocks were indeed sampled with replacement like in a bootstrapping procedure. We will make our language clearer in these lines to avoid confusion.
> - *sliding window line 188*
>     - The sliding window size is a parameter of the Sub2Vec-DBON method that is similar to the size of the context window used in skip-gram models. Given a random walk of a subgraph and a sliding window size of w, the subgraph embeddings are optimized to predict every set of w consecutive nodes in the random walk.
> - *citing work that uses encoding models for fMRI*
>     - Great point, we will include these. Some of these citations are more recent than our original manuscript which is why they are not mentioned, but we will update this section to reflect new work.
> - Bhattasli et al., 2018 is quite relevant and we will include it.

---

> > ### Comment · Reviewer_zeMS · 2021-08-27
> > **Weighing in on the syntax-semantics debate**
> >
> > Firstly, are there any voxels whose performance doesn’t improve on using BERT? If yes, then I agree with the reviewers that it is hard to disentangle if the lack of performance gain is because they _both_ encode syntactic information that is represented by the voxel (syntactic account) or semantic information that is encoded by the voxel (semantic account).
> >
> > However, since BERT has access to far less syntactic structure than ConTreGE and we know (anecdotally and here) that POSTags etc. barely explain any variance in the brain, I would bet that its the semantic overlap. Consequently, this would suggest that there are no exclusive “syntactic” region in the cortex. (See also work on the “semantic system” by Binder et al., 2009; Mitchell et al., 2008; Huth et al., 2016 etc.)
> >
> > Overall, I believe the interpretations of the results presented here. Having said that, the paper could definitely benefit from restating the claims, assumptions etc. clearly and perhaps refraining from calling the features spaces ‘purely syntactic’.

---

> > > ### Author Response · Authors · 2021-08-29
> > > **Responses to additional concerns**
> > >
> > > Thank you for the comments! Here are our responses:
> > >
> > > 1. Regarding the analysis in Table 1: The syntactic information analysis is one way of determining what type of syntactic structure is encoded in the embeddings. It is also a sanity check on the embedding process - to make sure that enough syntactic information is encoded by the embeddings. While it is true that the predictions could be because of the embeddings encoding some other information entirely, we find it implausible because of the nature of the construction process that only makes use of constituency trees. Many previous works have used encoding models that make use of multidimensional embeddings and this argument could be applied to any of them too. We see this as an inherent risk of the setup but one that is unlikely to be problematic in our case, especially since we use encoding models in a more controlled setting than usual in which we control for other feature spaces. Note that the POS+DEP tags are highly predictive of low-level syntactic nodes. The improvement in prediction performance when we use the ConTreGE embeddings is after controlling for the predictions made using POS+DEP tags. Thus, we are in a sense, indirectly controlling for the low-level syntactic nodes. Hence, we feel that we can conclude that the additional performance is likely because of the information about the higher-level nodes contained in the ConTreGE Incomp vectors. We talk about the limited interpretability of our embeddings in the limitations section, but we will include this risk in that section.
> > >
> > >
> > > 2. Regarding the ConTreGE Incomp vectors:
> > >
> > > - We agree with the reviewer that it would be very difficult for the brain to predict constituents that far into the future. However, this prediction task should become easier as more words are read. These subtrees are constructed in an automated fashion, and we do not tweak them based on the feasibility of the prediction task given its subjective nature.
> > > - Yes, it is likely that the ConTreGE Incomp embeddings are smoother than the ConTreGE Comp embeddings. However, the smoothness of a feature space isn’t always correlated with its prediction performance. Consider the punctuation-based feature space. Although it is not smooth, it is a great predictor of brain activity. In the same vein, the InConTreGE embeddings are also likely to be smooth but we see weaker predictions when they are used. We cannot rule out the reviewer’s hypothesis although we find it unlikely. Thus, we can add this hypothesis as another explanation for the results and leave it to more comprehensive future work on predictive coding to determine the correct explanation.
> > > - The InConTreGE feature space is constructed to encode the different possible parse trees that can derive the words seen till then. These different parse trees try to encode uncertainty about the parent nodes of the words seen till then. They also do not encode any information about possible future children of these parent nodes.
> > > However, with ConTreGE Incomp, we encode parts of the **final** parse tree of a sentence. It does not encode any uncertainty. With this embedding space we are trying to measure the ability of the brain to correctly predict future constituents.
> > > We see uncertainty about the current parse and the ability to correctly predict future constituents as two related but distinct concepts. This is the reason why we use the ConTreGE Incomp to hypothesize about the brain encoding correct future information.
> > >
> > >
> > > 3.  We only consider the productions used in the final parse tree of the sentence and do not take into account considerations about how constituents could have been completed in other contexts. Since the subtree beginning at VP also has other words as leaves, it is not considered complete until those words are seen.
> > >
> > >
> > > 4.  We thank the reviewer for their perspective on syntax-semantics debate. Like we point out in Figure 4, we see improvements in prediction performance in every language ROI when BERT is added. Although there may be a few sparse voxels in the language system in which we don’t see improvements, given the noisiness of the fMRI data, it would be hard to conclude that these voxels exclusively encode syntax. Thus, we base our conclusions on the aggregated ROI-level metrics. We will take into consideration the points put forth by the reviewers regarding this debate and modify our language in the final version of the manuscript.

---

> > > > ### Comment · Reviewer_zeMS · 2021-09-01
> > > > **Thank you for responses**
> > > >
> > > > Thank you for clarifying my questions above! I am fairly convinced about the arguments made for points 2 & 3 but would caution the authors to change the wording in the paper to reflect this:
> > > > - largest subtree that is "complete" for the given sentence
> > > > - adding the description they've provided here about why ConTreGE Incomp encodes future information and how it differs from InConTreGE (+ possible hypotheses)
> > > > - adding the parses for the longer sentence you provided here for clarification
> > > > - clarifications for the syntax-semantics debate and predictive coding claims etc.
> > > >
> > > > Re syntax-semantics: I'm glad and unsurprised that no voxels exist where BERT doesn't help. I think this more strongly speaks to your claims that there are no distinct regions for syntax.

---

> > > > > ### Author Response · Authors · 2021-09-01
> > > > > **Thank you for the suggestions and comments!**
> > > > >
> > > > > Thank you for the suggestions and comments! We will make these changes in the final version of the paper and include the additional parses in the supplementary material.

---

> > ### Comment · Reviewer_zeMS · 2021-08-27
> > **Additional concerns and responses**
> >
> > Thank you for the clarifications! Here are some additional questions:
> >
> > 1. After re-evaluting the paper, I am not convinced the analysis in Table 1 is demonstrative of anything. Sure showing that the POS+DEP Tags or the ConTreGE vectors can predict high-level nodes implies they preserve some information about the structure of the tree. But how do we know that’s the feature corresponding to high prediction in a voxel? In a similar vein as reviewer KgNW’s comments, the model could be relying on different types of information encoded in the feature spaces. Without ablating this information, decoding it etc., it is hard to claim that the brain does X (not to say that the statement “this suggests that the brain represents complex hierarchical syntactic information such as phrase and clause structure” is surprising)
> >
> > I just think this phrasing is misleading and would be interested in understanding what this experiment demonstrates beyond the claims stated above.
> >
> > 2. On looking at the example trees in the drive document and other reviewer’s comments, I also believe that stating ConTreGe Incomp to be suggestive of “predicts future structure from structure it already knows” is contentious at best. Consider example 1 on Pg-3 of the drive doc that shows an Income tree for “Harry”. Firstly, why would we assume the full tree likely has a conjunction and another CP? Under this construction, the incomplete trees for  “Harry had never believed” are far more similar to each other than the complete trees per words.  Thus, a valid alternative hypothesis to `Incomp` doing better than `comp` would be that the 4 words per TR have more similar features under this space, leading to temporally smooth downsampeld features that are just better correlated with the smooth BOLD.
> >
> > I think the Incremental ConTreGE vectors are far more representative of “predicting” future structure.
> >
> > 3. This is minor, but under the definition provided here, shouldn't the subtree for believed should be VP (largest tree) since VPs can take empty complements?

---

### Official Review · Reviewer_djyf · 2021-07-15

**Rating:** 7
**Confidence:** 3

**Summary:**

The paper proposes a new set of syntactic tree embeddings, and uses them capture additional variance in fMRI data beyond simpler complexity metrics. I think the paper setup is compelling overall, it is well-situated relative to prior work, and the results are intriguing. My primary concern is that the paper doesn't conclusively establish that these new embeddings are purely syntactic and disentangled from either complexity metrics or (more ambitiously) anything that BERT can capture, and this damages the downstream claims about neural correlates of syntactic representations.

I hope this can be addressed and solved in rebuttal/revision, since I find the paper good otherwise and hope to see it at the conference.

EDITED: I think the authors have reasonably addressed my concerns, and am raising my rating accordingly.

**Ethical Concerns:**

No concerns.

**Limitations And Societal Impact:**

The discussion of limitations is good, though missing the concern above regarding semantic leakage.

**Main Review:**

# Are these embeddings truly disentangled syntax embeddings?
The paper makes strong claims that its embeddings need to be purely syntactic (both in the introduction starting l99 and the discussion starting l340). While much care is taken to ensure that semantics or other language features do not "leak" into the embedding explicitly via e.g. a lexicalized grammar, the paper does not provide an empirical demonstration of disentanglement / orthogonality. This is important in light of the paper's framing because the embeddings could nonetheless be "incidentally" correlated with other features, either by random chance or more realistically if there is unavoidable correlation between structures and semantics (e.g. longer sentences are used with some words/concepts but not others, which doesn't seem unreasonable). The paper should do more to demonstrate that these embeddings are truly "purely" syntactic (e.g. showing that on they are empirically orthogonal to the other embeddings), or moderate its claims. This is particularly important given that the distributed significance maps could equally well imply a distributed representation for syntax as a leak of other information into the embedding.

To make room for this additional demonstration, I think the introduction containing extended background work could be shortened. Even if the extended background is kept, I would recommend writing a short introduction that puts the paper's contribution front-and-center, as it is currently well into the third page of the paper. A longer background discussion can follow. In addition, I think the section beginning l285, which seems to be a sanity check of the new embeddings' ability to capture  structural information without addressing the disentangling issue above, could be moved to the appendix if additional room is needed.

# Other comments

- Surprisal may be unfamiliar to the broad NeurIPS audience, the paper may consider defining it if space permits.
- I think the complicated acronyms like ConTreGE do not much to remind the reader about the content of the embedding, and are hard to match to the simpler acronyms like CC, CI, INC used in the figures.
- The sliding window length for the embedding is first mentioned on line 188 but implicitly mentioned without definition on line 163.
- Why do the semantic and syntactic feature dimensionalities need to be matched (l213)?
- l214 has a stray x.


**Time Spent Reviewing:**

4

---

> ### Author Response · Authors · 2021-08-10
> **Response to Reviewer djyf**
>
> We thank the reviewer for their comments. The main issue raised by the reviewer is regarding the statement that the ConTreGE vectors are disentangled from both the simpler syntactic features as well as the BERT features. However, we do not make this claim. In fact, we create our iterative addition of a feature space approach in order to directly control for the shared information in a stepwise way (l263-274). Using this approach, we are able to capture the unique variance explained by ConTreGE after controlling for other syntactic features. Then, since we control for most syntactic information, we consider the unique variance explained by BERT to be due to semantic information.
>
> Further, as mentioned by the reviewer, we build the ConTreGE vectors out of constituency trees that do not contain the word identities as leaves. Instead, words are replaced by numbers from 1 to n (total number of words in the text presented). This is done to minimize the amount of semantic information. Parse trees are some of the most fundamental sources of syntactic information. We agree with the reviewer that there may remain some correlations between these trees and semantic information because of our usage of natural text. However, such correlation between semantics and syntax is an unremovable part of natural language. Since the brain processes such natural material every day, such correlations could very well be integrated into how the brain processes these types of information (i.e. the brain could be learning these correlations and processing syntax or semantic information using them).
>
> Finally, in order to empirically show that our ConTreGE vectors do not encode a significant amount of semantics, we run a simple analysis similar to the one used by Caucheteux et al. (2021). Like in Caucheteux et al. (2021), we try to predict a GloVe-based semantic vector (extracted using spaCy) using the BERT embeddings and the ConTreGE vectors. These predictions were performed using a simple ridge regression model whose regularization parameters were chosen based on leave-one-out cross-validation $R^2$ scores (performed using sklearn’s RidgeCV module). In order to get the final $R^2$ scores, we perform 10-fold cross validation with a separate RidgeCV model being fit for each train-test split. We find that the BERT embeddings achieve average $R^2$ scores of 0.185 (the $R^2$ scores are first averaged across all 300 dimensions of the GloVe-based semantic vectors and then across all 10 cross-validation splits). However, the graph embeddings achieve significantly lower $R^2$ scores with ConTreGE Comp, ConTreGE Incomp and InConTreGE achieving average $R^2$ scores of 0.052, 0.020 and 0.021 respectively. These scores are markedly lower than the ones obtained using BERT, illustrating that the graph embeddings contain very low amounts of semantic information. On the other hand, our syntactic information analysis (Table 1), shows that these embeddings do indeed capture a lot of syntactic information. We will make space in the paper to discuss the comments above and to explain this analysis.
>
> Other comments:
> - We will define surprisal if space permits
> - We will change the acronyms in the final version
> - We are not too sure what is meant in the third bullet point. The summation on line 163 is over the top 5 most probable partial parses output by the top-down parser.
> - Given that the dimensionality of BERT embeddings is 1024 and the number of time points is only 1291, we reduce the dimensionality of the BERT embeddings to reduce overfitting. Additionally, we chose to retain the same number of dimensions as the ConTreGE vectors so as to make fair comparisons between the predictions obtained using syntax and semantics. We can present the results obtained using more dimensions in the final version if all reviewers prefer it.
> - Will fix line 214
>
> References
>
> Caucheteux, Charlotte, Alexandre Gramfort, and Jean-Remi King. "Disentangling syntax and semantics in the brain with deep networks." International Conference on Machine Learning. PMLR, 2021.

---

> > ### Comment · Reviewer_djyf · 2021-08-22
> > **Thank you for the comments and additional analysis.**
> >
> > I agree that the paper does not claim to disentangle relative to BERT -- I only meant to say that uncovering structural information not captured by black box learned representations would make a substantially stronger paper. I don't think achieving this is necessary to clear the NeurIPS bar.
> >
> > My bigger concern was regarding the "purely syntactic" embeddings that don't leak any semantics, something that seems shared with reviewer KgNW. The authors acknowledge in their response that perfect disentangling may not be possible by the nature of human language itself, which also seems reasonable (and also consistent with my point about unavoidable correlations). They add a simple analysis to empirically demonstrate that these new embeddings are minimally correlated with GloVe embeddings, which I think is a realistic way to address the semantic leakage concern.
> >
> > My second concern, also shared with the same reviewer, was clarity and in particular the heavy introduction that buries the key contribution too deep in the main paper. The authors promised to address this as well in their response to the other reviewer. My minor points are likewise addressed (though it wouldn't be bad to add some note to the paper regarding the dimensionality question if space permits, rather than just have it in the review response).
> >
> > (Regarding my paint about lines 188 and 163: I'm not sure what I was thinking there, honestly. I must have mis-parsed lines 163-165 to somehow think that the sum was over words, but even that wouldn't explain how I got from there to my comment. I apologize).
> >
> > Based on the additional analysis and the proposed restructure, I will happily raise my rating. I wonder if reviewers KgNW and yLj7 think their comments were likewise sufficiently addressed? Given the ratings on both sides of the acceptance margin, it'd be great to arrive at consensus on whether the paper clears the bar.

---

> > > ### Author Response · Authors · 2021-08-26
> > > **Thank you!**
> > >
> > > Thank you for increasing your rating! We'll be sure to address your concerns and include the additional analysis in the final version.

---

### Official Review · Reviewer_yLj7 · 2021-07-16

**Rating:** 6
**Confidence:** 2

**Summary:**

The manuscript entitled “Can fMRI reveal the representation of syntactic structure in the brain” proposed a word embedding scheme with the consideration for incomplete sentence (used to test how future sentence structure is represented in the brain) and partial parses (used to represent the multiple top-down paths of the brain in interpreting language syntaxes). Words embedded by the proposed schemes are then mapped to the corresponding fMRI signals (ensured by the experiment design where each word is strictly 0.5 second) through ridge regression. Voxels in the brain that are significantly correlated to certain combination of the syntax structures are studied in the results.

**Ethical Concerns:**

A rough estimation based on the word counts in Harry Potter and the Sorcerer’s Stone (>5000 words in each chapter) and the speed of reading (0.5s for each word), each subject will undergo a scan of >40 mins plus any localization and calibration scans. Normal fMRI scan usually takes 20 mins or less. The long scan time is certainly a burden to the subjects, although written informed consent and IRB are provided.

**Limitations And Societal Impact:**

The author mentioned difference in the paradigm (naturalistic stimuli vs. designed sentences / phrases) would lead to different localization results in the discussion section, which is totally reasonable. However, they did not provide convincing premise to use the designated naturalistic paradigm: a more complex task for the subjects, such as the one adopted in this work, would introduce more external factors and confounding, which will possibly lead to inconsistent false positive regions being identified. The author is suggested to provide more neuroscientific evidence for the possible functional pathways during a reading task, in order to better elucidate the findings.

A minor comment on the methodology: to the reviewer’s understanding, the ridge regression performed to map word features with fMRI signals is a 15*n->n model. The dimensionality of 15 might be too high for the ridge regression. There exist more powerful techniques to study the relationship between two sequences, where one of the sequences has higher dimensionality.

**Main Review:**

Originality: Using a more comprehensive and integrated scheme to model complex syntax structure is the main contribution and novelty of this work. Various intrinsic feature of the syntax structure (e.g., predictability in incomplete sentence) could be captured by the proposed method, which is a major improvement over the previous works.

Clarity: Method and result section of the proposed work is generally clearly written. With the help from the provided source code, the model should be partially replicable. Yet as the data and model are highly coupled in this work, it is difficult to fully test the model and evaluate it vs. other methods.
Section for InConTreGE (pp. 4, line 147-158) is confusing to understand, even with the help from Fig. 1: unlike a full sentence represented by a subtree, an incomplete subtree can have arbitrary number of forms (and the corresponding measurements such as node count). How exactly ConTreGE Incomp was constructed from this near-infinite composition of words? In addition, brain regions that are correlated with ConTreGE Incomp seemed too extensive and symmetric as in Fig. 4, which in contrast to common understanding of the high-order language processing unit in the brain (which might be true given the complex task involved in this study).

Significance: The proposed work would lead to superior representability for language-related functional neuroimaging analysis than traditional functional localization studies. In addition to the proposed paradigm, more specific language-processing researches can also utilize this framework for better modeling of functional neuroimaging data. The proposed work shall be significant to a highly specialized community (cognitive neuroscience in language study).

**Time Spent Reviewing:**

1.5

---

> ### Author Response · Authors · 2021-08-10
> **Response to Reviewer yLj7**
>
> We thank the reviewer for their comments.  It is true that the encoding model part of our pipeline is definitely specific to this experimental paradigm (naturalistic reading with a constant word rate). However, the embeddings we propose can be constructed for any text and can thus be adapted to other experiments. Moreover, we perform an apples to apples comparison of the various features. We acknowledge that one experiment's data are not adequate to firmly support our conclusions. But, the limits on publicly available, relevant fMRI data and the constraints on the length of the paper make it difficult for us to validate our results using another dataset. This is left for future work.
>
> Regarding the InConTreGE and ConTreGE Incomp embeddings:
> 1. The section the reviewer is referring to seems to be the section related to the ConTreGE Incomp vectors and not the InConTreGE vectors (InConTreGE vectors are explained in lines 159-165). We will improve the explanation of the ConTreGE Incomp vectors in the final version of the paper.
> 2. ConTreGE Incomp vectors were computed using the most probable full parse of the sentence constructed using the Self-Attentive Encoder-based parser [25], just like ConTreGE Comp. The difference between ConTreGE Comp and ConTreGE Incomp is that when the ConTreGE Incomp vectors are generated, we retain all of the PSG productions starting from the root of the full parse tree that are required to derive the words seen till then. However, any non-terminals that aren't needed to derive the words seen till then are NOT expanded. We have created this anonymous document that shows more examples of complete and incomplete trees for a complex sentence - https://drive.google.com/file/d/1TZgzJAtjNoJyeAfyDyGHJeZbYqFDknFP/view?usp=sharing. We hope this document can help in understanding the generation process.
> 3. The InConTreGE vectors were constructed using a different parser - the incremental top-down parser [27]. This parser is capable of outputting the top N most probable partial parses as and when a word is read. InConTreGE is the weighted average of the embeddings of the top 5 parses.
>
> We agree that the symmetry and extent of the results obtained using ConTreGE Incomp are not predicted by classical findings. However, this conforms with recent results such as those presented by Ev Fedorenko's group. The discussion section talks more about this debate and mentions the relevant literature.
>
> Regarding the use of the naturalistic language task - it is gradually becoming the paradigm of choice for many language researchers. For example, out of the papers we cite in our manuscript, Brennan et al [8] and others [9,10,12,44,45,46] also use naturalistic listening tasks to investigate syntactic processing by measuring word-by-word processing load. Even though story reading or listening involves multiple types of processing (low-level processing, word recognition, building sentences, inferring meaning and social reasoning, etc.), researchers are gradually learning more and more about how these processes are mapped to different brain areas. For example it has already been shown that the regions defined by Fedorenko et al. [39,40] as the language network (which we use in our manuscript as well) process word-level syntactic [17,18,19] and semantic features as well as sequence meaning [20,32,33]. These results have been replicated by more recent studies. Information beyond the sentence level is thought to recruit other brain areas that we use in everyday language [18, Speer et al. 2009].
> We will include this background in the main text.
>
> It is true that our ridge regression setup is not ideal. However, it is typical of this line of work [17,18,19,20,32,33], and encoding models for fMRI trained with ridge regression have become well studied and their robustness is established. Nonetheless, we find that other methods that can handle variables with different dimensions (e.g. centered kernel alignment (CKA) [Kornblith et al. 2019]) produce similar results to our findings, and therefore our findings appear to be robust to modeling choice.
>
> The total scan time from the original study this data is taken from [18] is 45 minutes, which should correspond to a total of around an hour for the entire session. This is in line with the typical duration of an fMRI scan. As is the case with other naturalistic experiments, the fact that participants are reading a real story makes it a much more engaging experience than classical controlled tasks.
>
> References:
>
> Speer, Nicole K., et al. "Reading stories activates neural representations of visual and motor experiences." Psychological science 20.8 (2009): 989-999.
>
> Kornblith, Simon, et al. "Similarity of neural network representations revisited." International Conference on Machine Learning. PMLR, 2019.

---

> > ### Author Response · Authors · 2021-09-01
> > **Have the reviewer's concerns been addressed?**
> >
> > As the discussion period is ending soon, it would be great if we could know whether the reviewer's concerns have been addressed by our response or if they had any additional questions. Thank you!

---

> > ### Comment · Reviewer_yLj7 · 2021-09-02
> > **The authors' response have clarified most of my concerns, although this work is limited to a highly specialized community**
> >
> > The reviewer is highly appreciated for the authors' detailed response. It would be definitely helpful to add the neuroscientific background of the task paradigm used in this work as elaborated in the responses into the revised version of this work. The reviewer understands that the focus of this paper is the algorithm for the extraction of syntactic features, rather than investigating the prediction task. Thus, as the authors have stated, a simple ridge regression method as widely applied by the community can be a reasonable (albeit not optimized) choice. The reviewer will maintain the current rating, mainly due to the fact that the proposed model is mainly interested to a highly specialized community (cognitive neuroscience in language study). The author's response of "... the embeddings we propose can be constructed for any text and can thus be adapted to other experiments" confirms with the (potential) generalizability of the proposed model, however the statement did not provide new evidence for its value to a broader audience.

---

> > > ### Author Response · Authors · 2021-09-02
> > > **Relevance to the NeurIPS community**
> > >
> > > We thank the reviewer for their response. Although we agree that this work makes a cognitive neuroscience of language contribution, we do not think that it is only relevant for a highly specialized community. In recent years, many papers have been published on the cognitive neuroscience of language at NeurIPS and other similar ML venues (some of them are listed below). Moreover, neuroscience is one of the main subject areas at NeurIPS and Language for Cognitive Science and Brain Mapping are subareas. Thus, we think that this paper will be interesting to many attending the conference. This paper is also about the application of graph embedding methods for encoding parse trees - a topic of possible interest for those studying natural language processing. It is also relevant for anyone studying the brain’s representation of any evolving structure, a difficult and important problem in many areas of cognitive neuroscience, even outside of language.
> > >
> > > 1. Jain, Shailee, et al. "Interpretable multi-timescale models for predicting fMRI responses to continuous natural speech." Advances in Neural Information Processing Systems 33 (2020).
> > >
> > > 2. Toneva, Mariya, et al. "Modeling Task Effects on Meaning Representation in the Brain via Zero-Shot MEG Prediction." Advances in Neural Information Processing Systems 33 (2020).
> > >
> > > 3. Toneva, Mariya, and Leila Wehbe. "Interpreting and improving natural-language processing (in machines) with natural language-processing (in the brain)." Advances in Neural Information Processing Systems 32 (2019): 14954-14964.
> > >
> > > 4. Schwartz, Dan, Mariya Toneva, and Leila Wehbe. "Inducing brain-relevant bias in natural language processing models." Advances in Neural Information Processing Systems 32 (2019): 14123-14133.
> > >
> > > 5. Jain, Shailee, and Alexander G. Huth. "Incorporating context into language encoding models for fMRI." Proceedings of the 32nd International Conference on Neural Information Processing Systems. 2018.
> > >
> > > 6. Caucheteux, Charlotte, Alexandre Gramfort, and Jean-Remi King. "Disentangling syntax and semantics in the brain with deep networks." International Conference on Machine Learning. PMLR, 2021.

---

### Decision · Program_Chairs · 2021-09-27

**Decision:**

Accept (Poster)

**Comment:**

This paper spurred a lot of discussion and back and forth with the authors, leading two reviewers to raise their scores. One reviewer thought the work is of limited interest and focused on a small community. I do not think this is grounds for reducing its score, since neuroscience is a main area in NeurIPS, definitely historically but also nowadays.

A major issue was that the reviewers questioned whether the proposed embeddings are truly disentangled syntax embeddings. This point was partly addressed by the authors providing more analysis. However, the reviewers still would like to see a more careful and nuanced discussion of this issue, how to interpret the results in light of the syntax-semantics debate, and to make sure that limitations of the present work are discussed.

The reviewers also mentioned problems with clarity and paper organizations -- specific questions were mostly answered, but paper organization should be improved in the next revision.

Given the discussion, I believe the paper passes the bar for acceptance. My low confidence here is mostly because my insufficient familiarity with neurolinguistics.